# Learning to Manipulate Anywhere:
# A Visual Generalizable Framework For
# Reinforcement Learning

**Zhecheng Yuan**[1,5,6*],    **Tianming Wei**[2,5*],    **Shuiqi Cheng**[3],    **Gu Zhang**[1,2,5],
**Yuanpei Chen**[4],    **Huazhe Xu**[1,5,6]

[1] Tsinghua University IIIS, [2] Shanghai Jiao Tong University, [3] The University of Hong Kong,
[4] Peking University, [5] Shanghai Qi Zhi Institute, [6] Shanghai AI Lab
yuanzc23@mails.tsinghua.edu.cn, huazhe_xu@mail.tsinghua.edu.cn

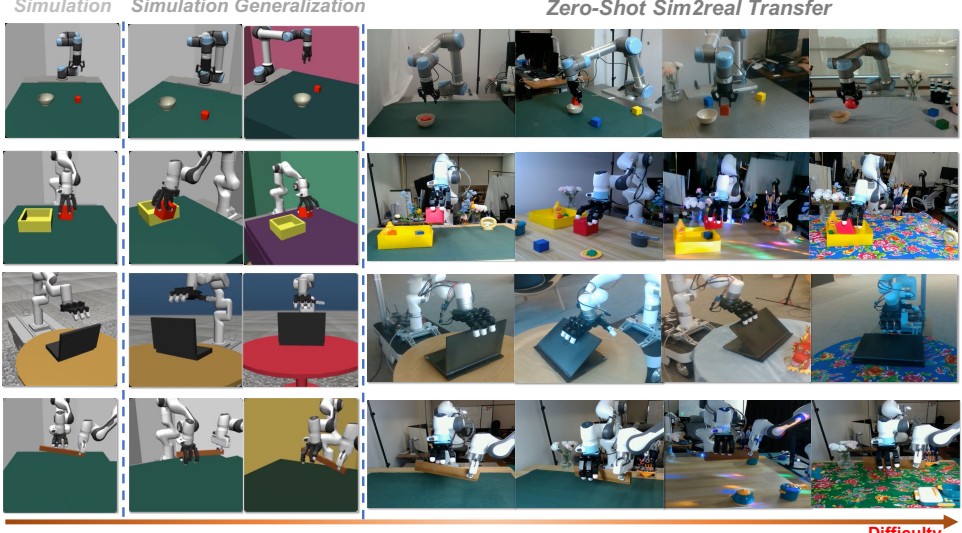

Figure 1: **Maniwhere.** Our framework is capable of training visuomotor robots that generalize effectively across various types of visual changes. Furthermore, Maniwhere can adeptly handle diverse real-world visual scenarios with various appearances and camera views in a zero-shot manner.

**Abstract:** Can we endow visuomotor robots with generalization capabilities to operate in diverse open-world scenarios? In this paper, we propose Maniwhere, a generalizable framework tailored for visual reinforcement learning, enabling the trained robot policies to generalize across a combination of multiple visual disturbance types. Specifically, we introduce a multi-view representation learning approach fused with Spatial Transformer Network (STN) module to capture shared semantic information and correspondences among different viewpoints. In addition, we employ a curriculum-based randomization and augmentation approach to stabilize the RL training process and strengthen the visual generalization ability. To exhibit the effectiveness of Maniwhere, we meticulously design **8** tasks encompassing articulate objects, bi-manual, and dexterous hand manipulation tasks, demonstrating Maniwhere's strong visual generalization and sim2real transfer abilities across **3** hardware platforms. Our experiments show that Maniwhere significantly outperforms existing state-of-the-art methods. Videos are provided at https://maniwhere.github.io/.

**Keywords:** Visual Generalization, Sim2real, Reinforcement Learning

---

*The first two authors contributed equally

8th Conference on Robot Learning (CoRL 2024), Munich, Germany.

# 1   Introduction

Visuomotor control tasks present roboticists with a vexing issue — the hardware setup can severely influence the performance of the robot policies. A prime example arises from the issue of immovable cameras - envision a carefully calibrated visual sensor, painstakingly positioned to enable seamless real-world deployment, only to have it disturbed by a lab mate. This single, seemingly innocuous incident can grind progress to a halt, forcing tedious recalibration or the collection of new demonstration data. Furthermore, changes in the background or the presence of extraneous objects within the captured views may undermine the effectiveness of a trained policy. Such obstacles have long plagued the field of robotics, representing critical barriers to realizing the full potential of advanced visuomotor systems.

Acknowledging these obstacles, when attempting to achieve sim2real visual policy transfer, it is common to instantiate a digital twin that closely resembles the actual real-world environment [1, 2, 3, 4, 5, 6, 7, 8]. Otherwise, the significant discrepancy between the digital twin and the real setting would render the trained models wholly ineffective. Hence, robots that adeptly handle in-the-wild scenarios should possess generalizability against various visual changes such as camera views, visual appearances, lighting conditions, etc.

While prior works have sought to tackle the challenges against visual scene variations [9, 10, 11, 12, 13, 14, 15, 16], these studies primarily focus on resolving a single aspect and are unable to handle multiple visual generalization types simultaneously. Meanwhile, it is non-trivial to incorporate various inductive biases into the training process. Naively applying domain randomization or data augmentation methods can destabilize the entire RL training, ultimately leading to divergence for the learned policy [4, 9, 12, 17]. More importantly, the generalization abilities of these methods have yet to be thoroughly evaluated on real robots.

In this paper, we are dedicated to enabling robots to acquire strong visual generalization ability so that they can step out of simulations and apply their learned skills to complex real-world scenarios without camera calibration. We introduce **Maniwhere**: A Visual Generalizable Framework for Reinforcement Learning. As shown in Figure 1, Maniwhere employs a multi-view representation objective to capture implicitly shared semantic information and correspondences across different viewpoints. In addition, we fuse the STN module [18] within the visual encoder to further enhance the robot's robustness to view changes. Subsequently, to achieve sim2real transfer, we utilize a curriculum-based domain randomization approach to stabilize RL training and prevent divergence. The resulting trained policy can be transferred to real-world environments in a zero-shot manner.

To conduct the evaluation, we develop **3** types of robotic arms and **2** types of robotic hands to design a total of **8** diverse tasks, alongside **3** corresponding hardware setups to validate the efficacy of our algorithm. Our comprehensive experiments demonstrate that, in both simulation and real-world scenarios, Maniwhere significantly outperforms existing state-of-the-art baselines by a large margin.

# 2   Method

In this section, we present Maniwhere, a generalizable framework for visual reinforcement learning. We propose a multi-view representation learning objective aimed at empowering the training agent with the ability to extract invariant features and generalize across different viewpoints. To further augment the model's spatial awareness, we incorporate an STN module into the visual encoder by actively spatially transforming feature maps. Additionally, we employ a curriculum of domain randomization to stabilize reinforcement learning (RL) training and facilitate sim2real. Next, having established the blueprint for Maniwhere, we proceed to elaborate it with details.

## 2.1   Multi-View Representation Objective

To endow the agents' ability to adapt to different viewpoints, we propose a multi-view representation learning objective to achieve this property. At each timestep $t$, the simulation returns the RGBD

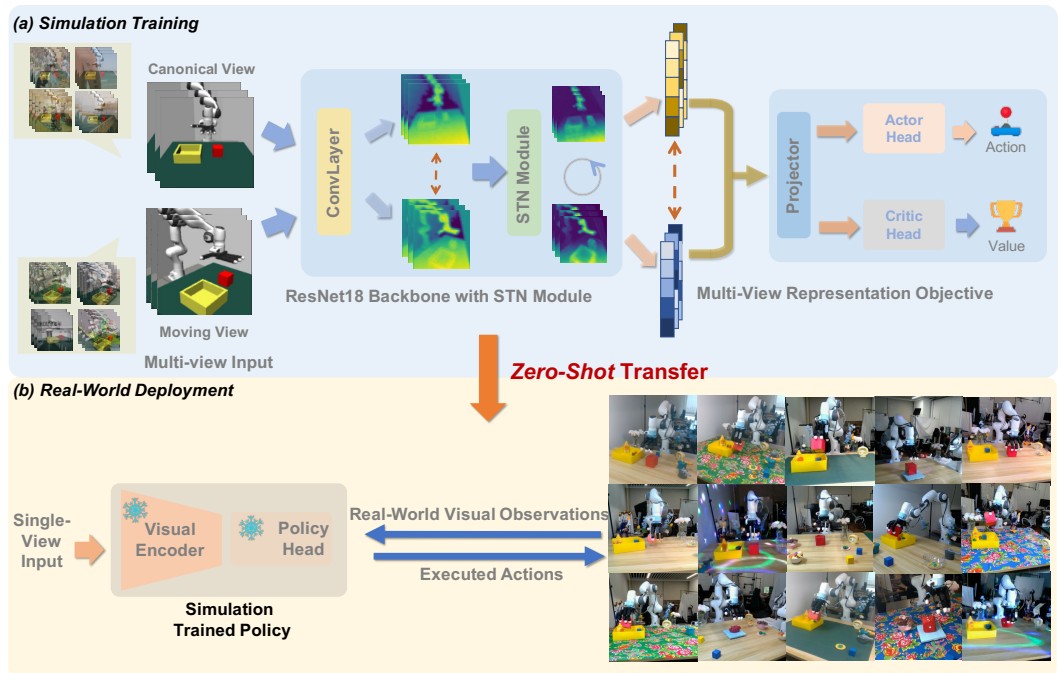

Figure 2: **Overview of Maniwhere.** The agent takes two images as input captured from different viewpoints with data augmentation and then passes them through a visual encoder containing an STN module to obtain visual representations. Subsequently, we employ multi-view representation learning to train the visual encoder while using a curriculum learning approach to stabilize the entire RL training process. Once the agent is trained in simulation, we can perform sim2real transfer.

observations from two cameras with different views: one is fixed, and the other one will randomly appear at different viewpoints. The range of camera randomization are listed in Appendix B.1. We denote the observation from the fixed viewpoint as $\mathbf{o}^{fixed}$, the observation from the randomized one as $\mathbf{o}^{move}$, and the visual encoder as $f_\theta$, which is parameterized by $\theta$. With respect to the first term, we adopt InfoNCE [19] to formulate our contrastive loss function $J_{con}(\theta)$:

$$J_{con}(\theta) = -log\frac{exp(f(\mathbf{o}^{fixed})^T \cdot f(\mathbf{o}^{move+})/\tau)}{exp(f(\mathbf{o}^{fixed})^T \cdot f(\mathbf{o}^{move+})/\tau) + \sum_{move-} exp(f(\mathbf{o}^{fixed})^T \cdot f(\mathbf{o}^{move-})/\tau)} \tag{1}$$

Here $o^{move+}$ is the positive example of $o^{fixed}$, which is rendered at the same timestep, while $o^{move-}$ is the negative example at different timesteps from the same batch samples. Inspired by Moco-v3 [20], we utilize a symmetrized loss form to gain better performance.

Moreover, recent works [21, 22] find that feature maps can be utilized to indicate correspondences of the images that share similar semantics. Hence, to endow agents with the ability to learn correspondences between different views, we also introduce an alignment loss function applied to feature maps across various layers:

$$J_{feat}(\theta) = \sum_{(\mathbf{o}^{fixed}, \mathbf{o}^{move}) \in \mathcal{B}} \|\mathbf{F}(\mathbf{o}^{fixed}) - \mathbf{F}(\mathbf{o}^{move})\|_2^2 \tag{2}$$

where $\mathcal{B}$ is the sampled batch, $\mathbf{F}$ is the flattened feature map embedding from a certain layer. The overall objective of Maniwhere is formulated as follows:

$$\mathcal{L}_{Maniwhere}(\theta) = J_{con}(\theta) + \lambda J_{feat}(\theta) \tag{3}$$

where $\lambda$ is the coefficient to weigh the scale between two terms. Through the guidance of $\mathcal{L}_{Maniwhere}(\theta)$, the agent enables to gain a better understanding of the semantics, correspondence, and view-invariant information within the whole visual scenarios via multi-view images extraction.

## 2.2 Curriculum Domain Randomization

Due to the high sensitivity of RL training towards different types of randomization, introducing additional noise can potentially lead to divergence in the entire training process. However, domain randomization and augmentation are indispensable for the sim2real transferability. Therefore, we propose a curriculum randomization approach in which the magnitude of randomization parameters is incrementally increased as training progresses. We employ an exponential scheduler to adjust the incremental change of the parameters. Additionally, we also establish a curriculum for the objective of stabilizing Q-value training [9]:

$$\left\| Q_\theta \left( f_\theta(\mathrm{aug}\,(\mathbf{o}_t)), \mathbf{a}_t \right) - \left( r_t + \gamma \max_{\mathbf{a}'_t} Q_{\theta'}^{tgt}(f_\theta(\mathbf{o}_{t+1}), \mathbf{a}'_t) \right) \right\|_2^2 \tag{4}$$

where $\mathrm{aug}$ is the augmentation method for the image observations, $Q_{\theta'}^{tgt}$ is the target $Q$ network. The augmented data incorporates increasing amounts of noise along with the training procedure. Here we choose SRM [11] with *random_overlay* [23], a frequency-based data augmentation as our augmentation method.

## 2.3 Inserting the STN Module

Spatial Transformer Network (STN [18]) enables the spatial transformation of data within the network, empowering the agent with enhanced abilities to perceive spatial information. Furthermore, to expand the model's capability for transformations beyond the 2D plane, we modify the affine transformations in the original STN to perspective transformations:

$$\begin{pmatrix} x_i^s \\ y_i^s \\ 1 \end{pmatrix} = \begin{bmatrix} \theta_{11} & \theta_{12} & \theta_{13} \\ \theta_{21} & \theta_{22} & \theta_{23} \\ \theta_{31} & \theta_{32} & \theta_{33} \end{bmatrix} \begin{pmatrix} x_i^t \\ y_i^t \\ 1 \end{pmatrix} \tag{5}$$

where $\theta_{ij}$ is the learnable transformation parameters, $(x_i^t, y_i^t)$ denotes the target coordinates on the output feature map's regular grid while the $(x_i^s, y_i^s)$ is the counterpart from the source image. Additionally, we leverage the first two layers of ResNet18 [24] as the backbone of visual encoder [13] and integrate the STN within it.

# 3 Experiments

In this section, we conduct numerous experiments in both simulated and real-world settings to showcase the effectiveness of Maniwhere in terms of generalizing to diverse visual scenarios with a combination of visual disturbance types.

## 3.1 Experiment Setup

**Tasks:** We have developed **8** tasks based on MuJoCo engine [25] with joint position control, including a variety of embodiments and objects such as single arm, bi-manual arms, dexterous hands, and articulated objects. We also establish the real-world counterparts for these tasks. In both simulation and real-world experiments, the observations are $128 \times 128$ RGB-D images with 3 frame stacks.

**Sim2real:** First, we train the agents in each simulated environment, where images from two different cameras will be observed: one offering a fixed viewpoint and the other moving throughout the given randomized range. Then, Maniwhere will integrate the knowledge from both viewpoints into the visual encoder via the approach mentioned in Section 2. Once finishing training in simulation, the

Table 1: **Generalization across different viewpoints.** The experiment result demonstrates that Maniwhere significantly outperforms the other baselines in all tasks with a $+\mathbf{68.5\%}$ boost on average.

| Setting | Method / Tasks | Lift Cube | Pick Cube To Bowl | Pull Drawer | Button with Dex |
|---------|----------------|-----------|-------------------|-------------|-----------------|
| | Maniwhere | $81.5_{\pm 7.0}$ | $89.5_{\pm 8.5}$ | $75.6_{\pm 9.2}$ | $97.6_{\pm 1.2}$ |
| | MV-MWM | $61.6_{\pm 22}$ | $9.6_{\pm 5.3}$ | $48.5_{\pm 21.1}$ | $77.6_{\pm 14.3}$ |
| | SGQN | $14.0_{\pm 8.2}$ | $2.8_{\pm 2.2}$ | $2.0_{\pm 1.0}$ | $12.8_{\pm 4.4}$ |
| | SRM | $14.8_{\pm 3.3}$ | $2.0_{\pm 2.8}$ | $3.2_{\pm 1.9}$ | $18.8_{\pm 4.1}$ |
| | MoVie | $10.5_{\pm 2.2}$ | $1.0_{\pm 2.3}$ | $5.0_{\pm 3.5}$ | $11.3_{\pm 4.7}$ |

| Setting | Method / Tasks | LiftCube Dex | PickPlace Dex | Close-Laptop Dex | HandOver Dex |
|---------|----------------|--------------|---------------|------------------|--------------|
| | Maniwhere | $88.8_{\pm 8.9}$ | $76.4_{\pm 9.2}$ | $82.4_{\pm 24.3}$ | $94.8_{\pm 4.8}$ |
| | MV-MWM | $78.0_{\pm 5.1}$ | $34.0_{\pm 28.9}$ | $69.5_{\pm 19.7}$ | $32.0_{\pm 23.1}$ |
| | SGQN | $14.0_{\pm 7.7}$ | $3.2_{\pm 4.6}$ | $15.0_{\pm 5.8}$ | $15.3_{\pm 4.6}$ |
| | SRM | $24.4_{\pm 8.0}$ | $6.4_{\pm 5.9}$ | $8.0_{\pm 4.2}$ | $16.0_{\pm 4.0}$ |
| | MoVie | $6.0_{\pm 2.2}$ | $1.0_{\pm 2.2}$ | $6.0_{\pm 4.1}$ | $7.0_{\pm 2.7}$ |

trained model will be directly transferred to the real world in a zero-shot manner. It should be noted that during both simulation and real-world evaluation, the trained agents receive images solely from a single camera for inference. The visual scenes will be modified from various aspects, including appearance, camera view, lighting conditions, and even cross embodiments at evaluation time.

**Real Robot Setup:** For gripper-based tasks, we utilize a UR5 arm equipped with a Robotiq gripper. Regarding tasks involving dexterous hand manipulation, we employ an Allegro Hand coupled with a Franka arm, and a Leap Hand [26] paired with an XArm mounted on a Ranger Mini 2 robot base from AgileX [27]. We use Realsense L515 camera to obtain visual inputs [28].

## 3.2 Baselines

We compare Maniwhere with the following visual RL leading algorithms: **SRM** [11]: implement a frequency-based augmentation method to achieve better generalization ability for visual appearances; **SGQN** [14]: SGQN leverages saliency maps to enhance the agent's attention on task-relevant information, and as suggested by Yuan et al. [23], it reveals better visual generalization capability across different camera views. **MoVie** [29]: utilizes domain adaptation to refine visual representations at new viewpoint through the dynamics model. **MV-MWM** [5]: applies MAE [30] to distill multi-view information into the visual encoder. It is worth noting that, unlike MV-MWM, Maniwhere does not require additional expert demonstrations, nor does it necessitate the acquisition of new data to adapt to environments as Movie does. Maniwhere can seamlessly transition to the real world in a zero-shot manner. We evaluate each algorithm over 5 seeds.

## 3.3 Simulation Results

**Generalize to different viewpoints.** In this section, we evaluate Maniwhere and the baseline methods across 8 challenging tasks. For each evaluation, 50 episodes from different viewpoints are tested. As shown in Table 1, compared to the existing baselines, Maniwhere achieves superior performance across all tasks with a large margin. The experiments indicate that the previous visual generalization algorithms struggle to manage visual changes in camera views. Regarding MoVie, while it adapts to the specific viewpoint change through domain adaptation, our setting involves different viewpoints among episodes. We find that MoVie cannot generalize to the visual scenarios where the viewpoint continuously changes. Hence, we argue that single-view image inputs are insufficient for fully perceiving spatial information. As for MV-MWM, it also utilizes multi-view images to enable the model to learn view-invariant features. Nevertheless, the experiment results exhibit that Maniwhere owns stronger multi-view generalization abilities than MV-MWM with a $+\mathbf{68.6\%}$ boost on average.

**Generalize to different visual appearances.** In addition to changes in camera views, we further alter visual appearances by perturbing the colors of the table and background. As shown in Figure 3, despite

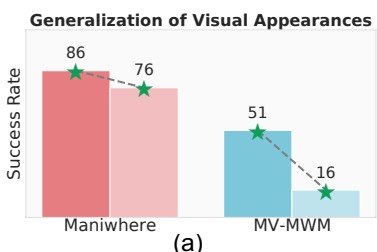
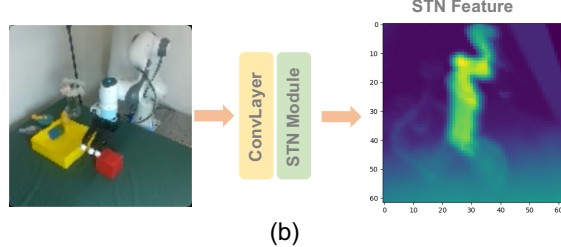

(a)  (b)

Figure 3: **(a). Generalization results of visual appearances.** Maniwhere exhibits minimal performance drop when encountering variations in visual appearance, whereas MV-MWM is unable to handle these visual scenarios. **(b). STN visualization.** STN is capable of transforming views from various other perspectives to align closely with the fixed view used during training.

the introduction of these visual appearance variations, Maniwhere maintains comparable performance levels with previous results, while MV-MWM suffers a substantial decline in performance drop. The underlying reason is that Maniwhere is compatible with various types of generalization, and our proposed objectives can effectively stabilize the impact of noise introduced by data augmentation and domain randomization.

**Generalize to different embodiments.** Then, we seek to validate the agent's generalization capability across different embodiments by replacing the UR5e robot arm with a Franka arm. As shown in Table 2, we surprisingly find that our trained model can directly perform zero-shot transfer to a different embodiment while maintaining the camera-view generalization ability. The qualitative analysis can be found in Appendix C.2.

| Task | Success Rate |
|------|--------------|
| LiftCube (ur5) | 82±7 % |
| LiftCube (franka) | 59±28 % |

Table 2: **The experiment results of Cross Embodiment.**

## 3.4 Real-World Experiments

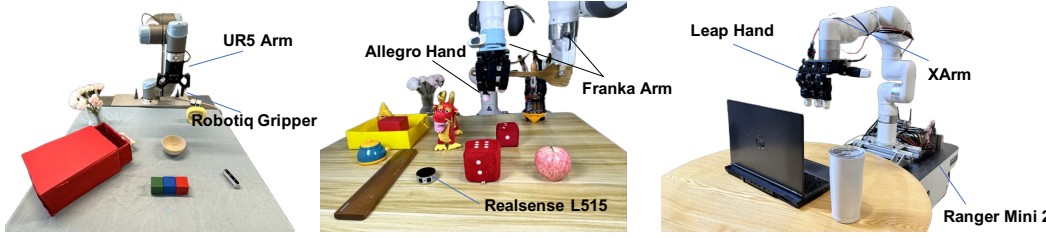

Figure 4: **Real-world setup.** Our real-world experiments encompass 3 types of robotic arms, 2 dexterous hands, and various tasks including articulated objects and bi-manual manipulation.

Regarding real-world experiments, as shown in Figure 4, we deploy our models trained in simulation on real-world scenarios across 3 hardware setups in a zero-shot manner. For gripper-based tasks, we implement multiprocessing alongside a shared memory queue to synchronize the execution of network inference and the controller [31], thereby ensuring a smooth movement process. As for dexterous-hand tasks, we introduce a moving average factor to reduce the jittering motions during execution [32, 33]. We select 5 challenging tasks in simulation to verify the effectiveness of agent's sim2real tranferability. For each task, we choose 5 different viewpoints that cover the workspace, and the visual appearances of the scenario will be altered under each viewpoint as well. Each algorithm is evaluated 5 trials under every visual condition. In each trial, yaw and pitch angles of the camera will

Table 3: **Real-world experiments.** Maniwhere outperforms MV-MWM with +**53.5%** on average.

| Method / Task | Drawer | LiftCube | Pickplace dex | CloseLaptop | Handover | **Average** |
|---------------|--------|----------|---------------|-------------|----------|-------------|
| MV-MWM | 2.0% | 12.0% | 0% | 20.0% | 2.0% | 7.2 ± 8.5 % |
| Maniwhere | **65.7%** | **78.0%** | **52.0%** | **72.0%** | **36.0%** | **60.7**±16.8% |

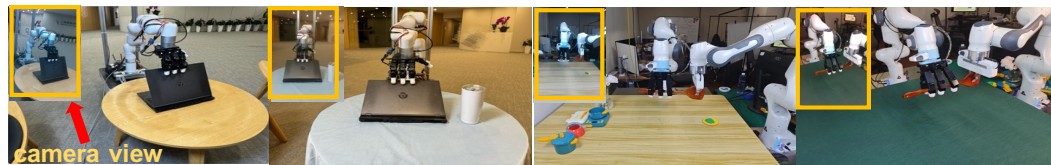

Figure 5: **Real-world snapshots.** Real-world experiments under different visual conditions.

be randomized. Figure 5 exhibits snapshots of real-world settings. As shown in Table 3, consistent with the simulation results, Maniwhere outperforms MV-MWM on all tasks. The experiment indicates that Maniwhere not only narrows the sim2real gap but also enables the trained robots to achieve the real-world generalization ability. More details and results can be found in Appendix and webpage.

**STN features.** Figure 3 (b) illustrates that when facing real-world images, the STN layer assists the agent in transforming inputs from different viewpoints to closely resemble the fixed view used during training, thus facilitating the camera-view generalization and acquiring view-invariant representations.

## 3.5 Ablations

To investigate the necessity of each component in Maniwhere, we ablate two main design choices in Maniwhere, including the multi-view contrastive representation learning objective and STN module. Our ablations are conducted on two lifting tasks and one pickplace task. As shown in Table 4, we observe that the multi-view objective contributed significantly to the improvement; without it, the model would be deprived of its ability to generalize across different camera views. Meanwhile, the integration of the STN

| Ablation | Success Rate |
|---|---|
| Maniwhere | $\mathbf{86.5}_{\pm \mathbf{3.9}}\%$ |
| w/o. multi-view objective | $29.4_{\pm 5.0}\%$ |
| w/o. STN | $65.3_{\pm 7.7}\%$ |
| w/ TCN | $65.8_{\pm 5.1}\%$ |

Table 4: **Experimental results showcasing various ablations.**

enhances the model's capacity to understand and adapt to spatial view changes. Furthermore, we adopt TCN loss [34], which also applies multi-view contrastive learning, to replace our multi-view objective. The results reveal that there remains a significant generalization performance gap compared with Maniwhere, highlighting the advantages of our approach.

## 3.6 Qualitative Analysis

To delve deeper into the reasons behind Maniwhere's superior performance, we examine it from the aspects of visual representations and Q-value functions of RL.

**Q-value distribution.** Conceptually, if an RL agent can produce the Q-value distribution from noisy visual inputs that closely approximates that obtained from the original images, the trained agent can be regarded as a more robust and generalizable learner. [12, 9] We visualize the representation of the penultimate layer of the critic using t-SNE to examine how the Q-distribution differs under various viewpoints with our proposed multi-view representation learning method. As shown in Figure 6, our method maintains a distribution similar to that of the original fixed viewpoint, whereas relying solely on the objective in Eq 4 fails to adapt to different camera views. Consequently, Maniwhere not only closes the distance between visual

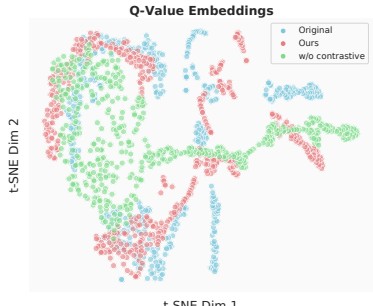

Figure 6: **Q-value embedding distribution.**

embeddings to obtain more robust visual representations, but also narrows the gap between Q-distributions, further stabilizing training and enhancing agent's visual generalization ability.

**Trajectory embedding.** For the visual representation side, we visualize the feature maps of images rendered from different viewpoints along the same execution trajectory, and then apply t-SNE to

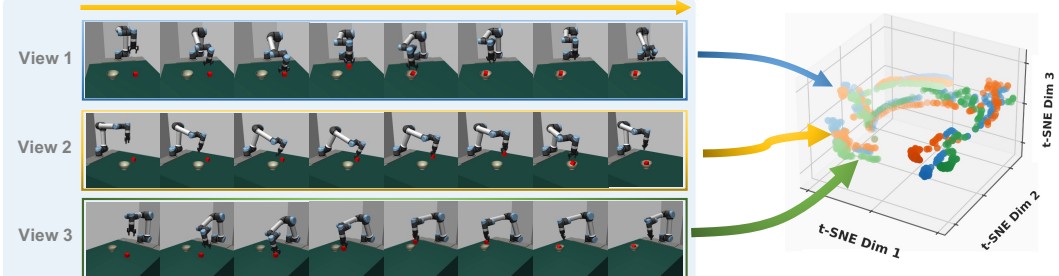

Figure 7: **Trajectory embedding visualization.** We capture images from 3 viewpoints at the same moment while executing an identical trajectory. As the timestep progresses, the color of the embedding becomes increasingly darker. We find that they exhibit similar visual representations.

embed the feature maps. Figure 7 shows that Maniwhere is capable of mapping the images from different viewpoints into the similar regions as well as maintain consistency throughout the entire execution trajectory.

## 4 Related Work

**Generalization in visual RL.** In recent years, multiple works have resorted to addressing the critical issue of generalization [6, 35, 36, 37, 38, 39, 40, 41, 42]. Based on the strong data augmentation, several studies integrate advanced methods such as pre-trained visual encoders [13], saliency maps [14], and normalization techniques [41] to enhance the visual generalization capabilities of agents. Despite these advancements, current methods primarily address only variations in visual appearances and fall short when confronted with other types of visual changes. Another line of works devote to solving the camera view changes. For instance, MoVie [29] utilizes an inverse dynamics model to facilitate the model adapt to a novel view pattern. However, it is limited to a singular type of view and cannot accommodate multiple different view patterns. Meanwhile, MV-MWM [5] leverages model-based RL to train a multi-view masked encoder. However, its dependency on demonstrations for task completion remains a significant limitation. Moreover, these two approaches are unable to adapt to the changes of visual appearances. On the contrary, Maniwhere offers a versatile visual RL approach that is compatible with multiple visual generalization types and does not require any demonstrations.

**Representation learning for visuomotor control.** Representation learning plays a critical role in visuomotor control tasks [43, 44, 45, 46, 47, 48, 49, 50, 51]. Recent works [13, 52, 53] have verified that leveraging the pre-trained visual encoders via representation learning approaches can facilitate the execution of numerous downstream control tasks. Furthermore, SODA [10] utilizes a BYOL-style objective to decouple augmentation from policy learning. RL3D [3] pretrains a deep voxel-based 3D autoencoder and continually finetunes the representation with in-domain data. H-index [54] applies the keypoint detection and pose estimation method to derive a customized representation for the hand. In contrast to these works, Maniwhere not only strives to obtain generalizable visual representations but also seeks to enable these representations to bridge the sim2real gap.

## 5 Conclusion and Limitations

In this paper, we present Maniwhere, a visual generalizable framework for reinforcement learning. Maniwhere leverages multi-view representation learning to acquire the view consistency information, and utilize curriculum randomization and augmentation approach to train generalizable visual RL agents. Our experiments demonstrate that Maniwhere can adapt to diverse visual scenarios and achieve sim2real transfer in a zero-shot manner. In the future, we plan to enhance Maniwhere's generalization ability across broader camera ranges and more diverse visual scenarios. Beyond visual generalization, we intend to incorporate spatial generalization methods to handle more complex object spatial relationships, with the ultimate goal of developing a robust sim2real framework.

**Acknowledgments**

We sincerely thank Yiping Zheng, Yanjie Ze and Yuanhang Zhang for helping set up the hardware. We also thank Jiacheng You, Sizhe Yang, and our labmates for their valuable discussions. This work is supported by the National Key R&D Program of China (2022ZD0161700).

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

# Appendix

## A  Task Description

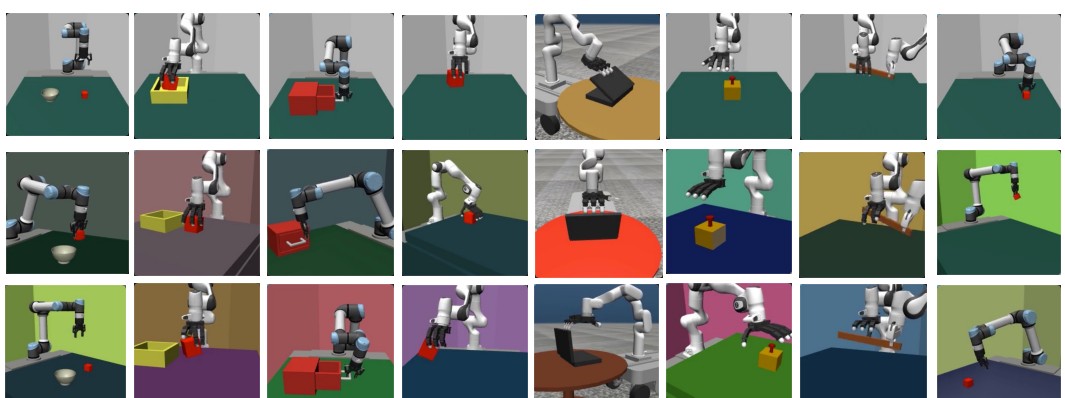

Figure 8: **Snapshot of all tasks and test visual scenarios.**

**Lift Cube:** This task involves a UR5 arm equipped with a Robotiq gripper. A red cube is placed on the table. The agents arerequired to grasp the cube and lift it off the table. A reward greater than 250 is considered a success. We lock 3 out of the 6 DoFs of the UR5 arm to restrict unnecessary movements and reduce the action space, facilitating more efficient RL learning.

**Pull Drawer:** This task contains a UR5 arm equipped with a Robotiq gripper. A drawer is placed on the table. The agents need to approach the handle and pull the drawer open. A reward greater than 230 is considered a success. We lock 3 out of the 6 DoFs of the UR5 arm.

**Pick Cube To Bowl:** Except for the red cube, we additionally place a bowl on the table. The agent needs to lift the cube and place it into the bowl. A reward greater than 230 is considered a success. We lock 3 out of the 6 DoFs of the UR5 arm.

**Button with Dex:** This task involves a Franka arm equipped with an Allegro Hand. The agent is required to press the button to receive the reward. A reward greater than 250 is considered a success. We lock 3 out of the 7 DoFs of the Franka arm and the DoFs of Allegro Hand.

**Close-Laptop Dex:** This task is equipped with a Leap Hand, an XArm, and a Ranger Mini 2 base from AgileX. The agent requires to close the laptop on the table. We lock the DoFs of Leap hand and 4 DoFs of Franka Arm. When the joint of the laptop is smaller than 1.7 rad, we consider it a success.

**LiftCube Dex:** This task involves a Franka arm equipped with an Allegro Hand. The agent is required to grasp the cube and lift it off the table. A reward greater than 50 is considered a success. We lock 3 out of the 7 DoFs of the Franka arm and use 4 DoFs of Allegro Hand (The rest of the DoFs will be set to a default value to keep a proper gesture).

**PickPlace Dex:** This task involves a Franka arm equipped with an Allegro Hand. The agent is required to grasp the cube and lift it off the table and place it to the box. A reward greater than 50 is considered a success. We lock 3 out of the 7 DoFs of the Franka arm and use 4 DoFs of Allegro Hand (The rest of DoFs will be set to a default value to keep a proper gesture). Additionally, we use the moving average technique to smooth the motion.

**Handover Dex:** We utilize two Franka arms, one equipped with a gripper and the other with an Allegro hand. This task requires cooperation between the two arms; the gripper must grasp a spatula and pass it to the hand. Success is determined if the distance between the hand and the object is less than 0.03 meters.

## B Implementation Details

### B.1 Environment Randomization Parameters

Table 5: Domain randomization parameters in Maniwhere.

| Attribute | Value |
| --- | --- |
| UR5 joint armature | $0.1 \cdot (1 \pm 0.1)$ kg m$^2$ |
| UR5 shoulder pan joint damping | $360 \cdot (1 \pm 0.1)$ N s/m |
| UR5 shoulder lift joint damping | $280 \cdot (1 \pm 0.1)$ N s/m |
| UR5 elbow joint damping | $250 \cdot (1 \pm 0.1)$ N s/m |
| UR5 wrist joint damping | $280 \cdot (1 \pm 0.1)$ N s/m |
| Franka joint armature | $0.1 \cdot (1 \pm 0.1)$ kg m$^2$ |
| Franka joint damping | $1 \cdot (1 \pm 0.1)$ N s/m |
| XArm joint damping | $15 \cdot (1 \pm 0.1)$ N s/m |
| XArm joint frictionloss | $4 \cdot (1 \pm 0.1)$ |
| Object cube size | $0.05 \cdot (1 \pm 0.1)$ m |
| Table height | $[-0.01, 0.01]$ m |
| Cube randomized range | $[0.6 \sim 0.8, -0.15 \sim 0.15]$ m |
| Dex cube randomized range | $[0.65 \sim 0.85, -0.1 \sim 0.11]$ m |
| Drawer randomized range | $[0.7 \sim 0.8, -0.3 \sim -0.2]$ m |
| Button randomized range | $[0.6 \sim 0.8, -0.15 \sim 0.15]$ m |
| Laptop randomized range | $[-0.125 \sim -0.075]$ m |
| Laptop angle randomized range | $[-0.15 \sim -0.05]$ rad |
| Camera looking target position in world frame | $[0.6, 0.0, 0.2]$ |
| Camera elevation angle | $[10.5, 30.5]°$ |
| Camera azimuth angle | $[-60, 60]°$ |
| Camera Fov | $[38, 46]°$ |
| Camera Distance | $[1.12, 1.54]$ m |
| Action-delay | $[0, 2]$ timesteps |
| Control timestep | $[0.016, 0.024]$ s |

### B.2 Curriculum Randomization

For each task, a threshold of $2e5$ steps is established as the initial frame for domain randomization. The randomization parameters will vary exponentially within the ranges specified in Table 5 starting from the $2e5$-step mark (the Close Laptop task beginning at $7e4$ step). Concurrently, the stabilizing objective described in Eq 4 will process augmented images from the fixed view prior to this threshold, and will incorporate augmented images from the moving view thereafter.

### B.3 Hyper-Parameters

We list the training hyper-parameters used in Maniwhere in Table 6.

## C Additional Results

### C.1 Real-world Experiments

**Real-world setup.** Due to the limitation that a single PC cannot control two Franka arms simultaneously, we developed a control logic framework using zmq to coordinate three PCs. In this setup,

Table 6: Hyper-parameters in Maniwhere.

| Hyper-parameters | Value |
|---|---|
| Input size | $128 \times 128$ |
| Discount factor $\gamma$ | 0.99 |
| Replay Buffer size | int(1e7) |
| Feature dim | 256 |
| Action repeat | 1 |
| N-step return | 3 |
| Optimizer | Adam |
| Frame stack | 3 |
| Temperature of InfoNCE | 0.1 |
| Learning Rate of STN | 1e-4 |
| $\lambda$ | 200 |

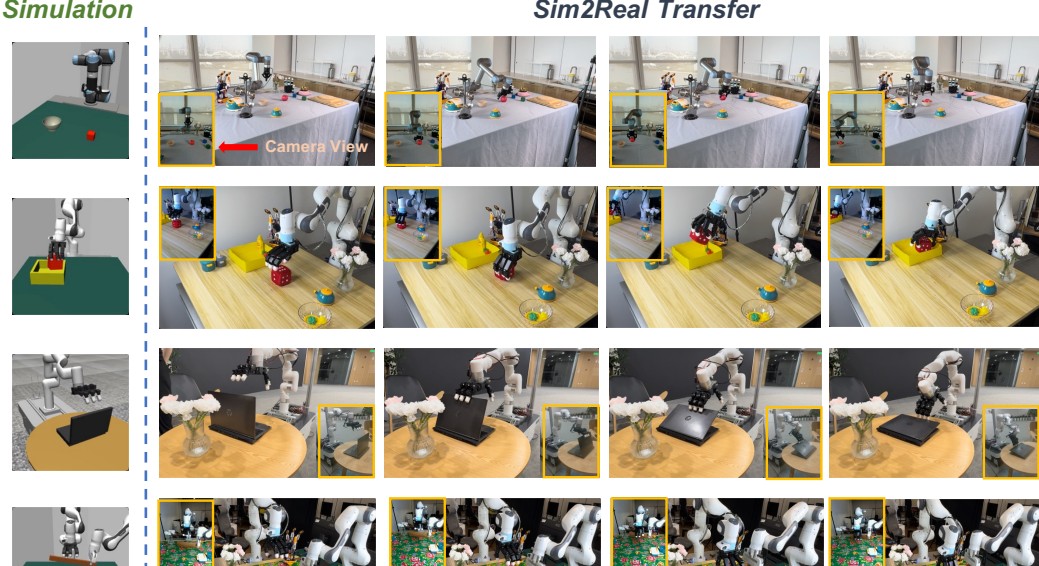

Figure 9: **More real-world snapshots..** We exhibit more real-world snapshots in challenging real-world visual scenarios.

one PC is regarded as the client, while the other two serve as servers. The client PC receives visual input and performs network inference, subsequently transmitting the inferred actions via socket connections to the two server PCs. The server PCs are responsible for controlling the Franka arms and executing the received actions. This process is iterative, with the servers sending new visual input back to the client for continuous processing. Given that MV-MWM has a large model size and requires substantial memory for loading, we deployed it on a desktop equipped with an RTX 3090 GPU. In contrast, the deployment of Maniwhere demands significantly less hardware, allowing it to perform inference even on CPU desktops. Regarding the camera setup, we establish the evaluation viewpoints at three yaw angular ranges: [0, 5°], [10, 25°], and [40, 55°], on both the left and right sides. Additionally, across the five trials conducted at each viewpoint, the camera height will be varied within a range of -3 to 3 cm.

**Instance generalization.** Thanks to the general grasping capabilities of the dexterous hand, Figure 10 shows that Maniwhere is not limited to a single object when executing the *lifting* behaviours and can generalize across different instances with various shapes and sizes.

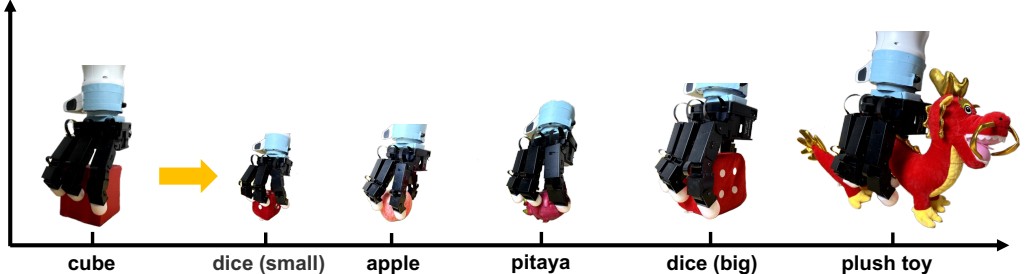

Figure 10: **Instance Generalization.** We find that Maniwhere won't overfit to the specific object size and shape.

## C.2 Cross Embodiment

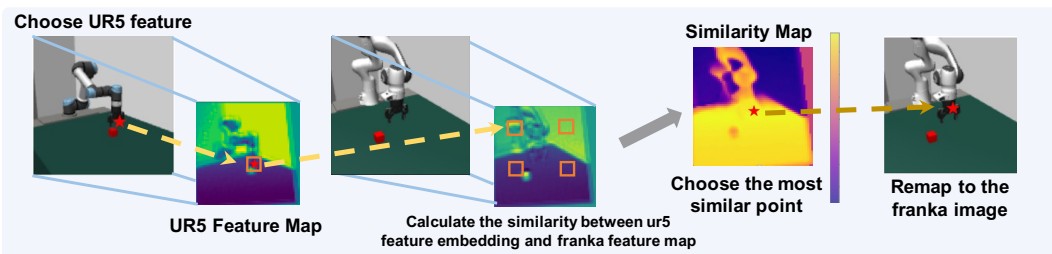

Figure 11: **Feature Correspondence.** Maniwhere can find the feature correspondence between different embodiments.

Figure 11 illustrates that when we first select a pixel point on the UR5 original image (marked with a red pentagram) and extract its feature (enclosed in the orange square) after passing through the convolutional layer, we compute its normalized cosine similarity with the image feature of Franka arm to obtain a similarity map. The point with the highest value in this map is identified as the most similar point between two images (marked with a red pentagram). As shown in Figure 11, Maniwhere can effectively recognize semantically consistent positions between the two different embodiments. With respect to randomization, to enable the agent to capture the correspondence information through the multi-view representation objective, we do not augment the moving view image in Eq 4.

## C.3 View Generalization

We further investigate how Maniwhere's performance varies across different camera view ranges. We divide the randomized camera view range into three parts, within each of which the camera's pitch and field of view are randomly altered as well. The value for each range is calculated as the average of both the left and right sides. Due to the excessive angular range in `handover` task potentially obscuring the other arm, we confined the range for this task to 0-30 degrees. Table 7 illustrates that, although Maniwhere's performance exhibits a slight decline as the angle increases, it still retains the capability to handle these scenarios effectively.

Table 7: **Generalization across different camera view ranges.** Maniwhere retains the generalization capability to handle these scenarios effectively. We evaluate 20 episodes in each range.

| Method / Task | LiftCube Dex | PickPlace | Pickplace dex | Button dex | Handover |
|---|---|---|---|---|---|
| range [0, 15]° | 91.3% | 91.0% | 82.5% | 97.5% | 94.0% |
| range [20, 35]° | 88.3% | 88.0% | 81.5% | 97.5% | 94.0% |
| range [45, 60]° | 86.9% | 84.0% | 65.0% | 94.4% | 92.0% |

## C.4 Depth information helps sim2real transfer

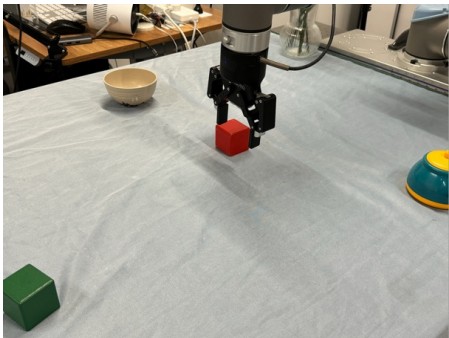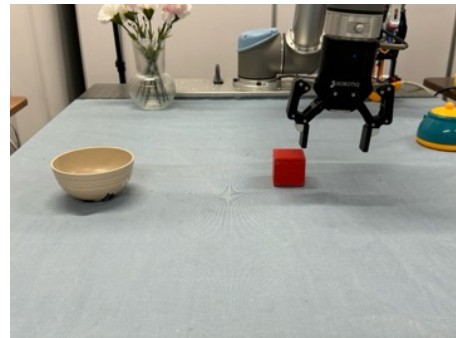

Figure 12: **Spatial illusion.** These two figures are captured at the same timestep. Without depth information, we lose the front-to-back positional relationship between the object and the gripper in the three-dimensional world.

To ensure the depth images closely resemble real-world conditions, we first pre-process the depth image. We introduce Gaussian noise $\mathcal{N}(0, 0.01)$ and depth-dependent noise $\mathcal{N}(0, depth\_scale)$, where the depth_scale equals np.abs(depth_image) * 0.05. Then, we apply GaussianBlur to smooth the noise. Additionally, the depth values are clipped to within 2 meters and normalized to the range [0, 255]. During sim2real, we find that depth image can largely help to alleviate the ambiguity situation. Figure 12 shows that when encountering large camera viewpoints, the agent cannot accurately determine the grasping position since RGB information alone does not provide the necessary front-to-back positional relationship between the object and the gripper in the 3D world. However, by incorporating depth images, we observe a significant improvement in real-world scenarios.

## C.5 MV-MWM with data augmentation

We also apply the data augmentation method on MV-MWM. As shown in Table 8, MV-MWM suffers a significant performance drop while facing data augmentation. These results are consistent with the recent works [9, 12].

| Task | Success Rate(w/o DA) | Success Rate (w/DA) |
|------|----------------------|---------------------|
| Button Dex | $77.6_{\pm14.2}$ % | $1.3_{\pm2.3}$ % |
| PickPlace Dex | $34.0_{\pm28.9}$ % | $8.7_{\pm13.3}$ % |

Table 8: **MV-MWM with data augmentation.**

Naively applying data augmentation can cause instability and large variance during training. In turn, the results also prove that simultaneously handling multiple types of generalization is non-trivial and highlights the superiority of Maniwhere.

## C.6 Regarding target object color

Although we found that the agent demonstrates strong generalization capabilities when the visual scene is altered, including changes to the table, background, and the introduction of colorful distractors, it fails the task when the color of the target object is changed. Figure 13 exhibits that during executing a trajectory, the agent focuses more attention on the target object while ignoring task-irrelevant information, making it more sensitive to changes in the color of the target object. We use the Grad-CAM [55] to visualize the agent's attention.

## C.7 The implementation of MV-MWM

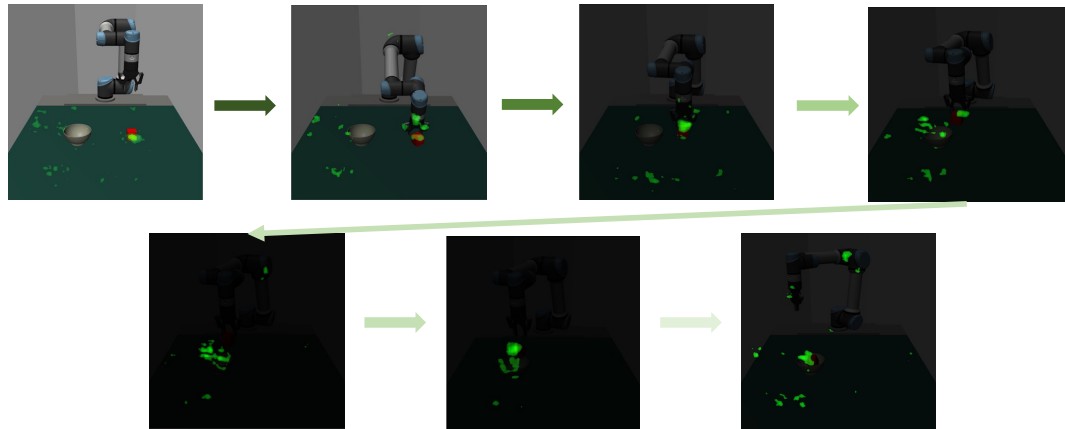

Figure 13: **Visualization of the agent's attention by Grad-CAM.**

We introduce additional design adjustments to tailor MV-MWM to our setting, enabling it to exhibit its potential performance. We utilize a trained agent as an expert to collect the same number of state-action pairs as in the original paper for pre-training, and ensure these pairs are from high-reward trajectories. Furthermore, consistent with Maniwhere, we employ RGBD input as the input modality. As shown in Table 9, when facing two third-person view images during training, the modified MV-MWM demonstrates stronger performance compared to its original version.

| Task | Original | Modified (Ours) |
|------|----------|-----------------|
| Lift Cube | $6.7_{\pm 2.9}$ % | $61.6_{\pm 22}$ % |
| Pull Drawer | $12.0_{\pm 8.4}$ % | $48.5_{\pm 21}$ % |
| Button Dex | $64.0_{\pm 19.8}$ % | $77.6_{\pm 14.3}$ % |

Table 9: **The performance of our modified MV-MWM.**

### C.8 The utilization of data augmentation

Effectively leveraging data augmentation is crucial for achieving visual appearance generalization. Existing approaches [12, 13, 9] have demonstrated that naively applying data augmentation can lead to training instability and divergence. To address this, we employ the objective outlined in Eq 4, which allows for the introduction of noise to enhance model robustness while simultaneously stabilizing Q-value training. Additionally, we integrate the frequency-based method [11] to further improve the model's generalization ability and narrow the sim2real gap. As shown in Table 10, without our data augmentation approach, the agents lack generalization capability in both simulation and real-world settings. Therefore, the data augmentation strategy utilized in Maniwhere proves to be effective in equipping the robots with the ability to handle visual appearance changes.

| Task | without DA | Ours |
|------|-----------|------|
| Lift Cube | $77.6_{\pm 14.2}$ % | $81.5_{\pm 7.0}$ % |
| Pull Drawer | $2.7_{\pm 1.1}$ % | $75.6_{\pm 9.2}$ % |

Table 10: **The effectiveness of data augmentation.**

### C.9 Imitation Learning

Beyond visual RL, we also conduct experiments in Imitation Learning (IL) to verify the effectiveness of Maniwhere. The *Pickplace* task with dex-hand is selected for evaluation. In this setting, we utilize the RL trained policy as the expert to collect 100 demonstrations, and apply Diffusion Policy [31] with RGBD input as the training algorithm. Consistent with RL, we use the same visual encoder and

| Method | Success Rate |
|--------|-------------|
| Maniwhere | $\mathbf{68.7_{\pm 2.3}}$% |
| Diffusion Policy | $10.7_{\pm 3.1}$ % |

Table 11: **The experiment results in Imitation Learning.**

proposed multi-view representation learning objective for training. As shown in Table 11, Maniwhere demonstrates robust generalization capability as well.

## C.10 The training curves

Figure 14 demonstrates the camera view generalization ability of Maniwhere during the whole training process. The curves are smoothed with window size 6.

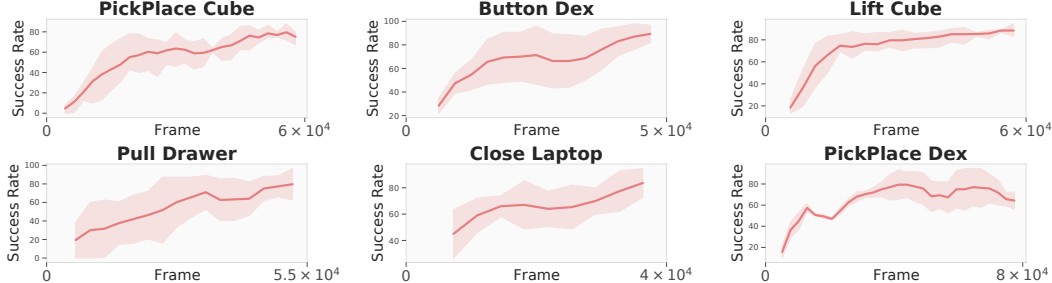

Figure 14: The training curves of Maniwhere across tasks.

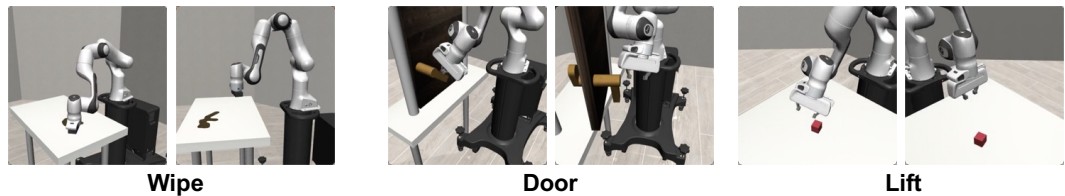

**Wipe**                    **Door**                    **Lift**

Figure 15: The visualization of 3 Robosuite tasks at diffrent viewpoints.

## C.11 Evaluation on Robosuite

In terms of reproducibility, We evaluate our method on the widely-used robotic benchmark robosuite [56] to verify the effectiveness of Maniwhere. We conduct 3 typical manipulation tasks in this benchmark. As shown in the Table 12, Maniwhere can also exhibit superior performance on three robosuite tasks. The three tasks visualization are shown in Figure 15.

| Task | Success Rate |
|------|--------------|
| Lift | $72.0 \pm 15.2\%$ |
| Door | $95.2 \pm 8.7\%$ |
| Wipe | $75.0 \pm 6.2\%$ |

Table 12: **The experiment on Robosuite.**

