# OpenReview forum: "Learning to Manipulate Anywhere: A Visual Generalizable Framework For Reinforcement Learning"
_robot-learning.org/CoRL/2024/Conference — CoRL 2024_

### Official Review · Reviewer_vrac · 2024-07-20
**Exciting work on sim-to-real manipulation while being entirely background and viewpoint invariant**

**Originality:** 5
**Technical Quality:** 4
**Clarity Of Presentation:** 4
**Potential Impact:** 4
**Recommendation:** 4
**Confidence:** 5

**Review:**

## Summary
In this work, the authors present a class of general policies that can solve manipulation tasks in a background and viewpoint invariant fashion. The core of the work is using simulation environments to randomize the background and camera viewpoints, while also using a STN module on the visual encoder to encourage viewpoint independence.

The training process has two objectives, one is for multi-view agreement in the representation space, and the other is the standard RL objective. Using this, the authors present trained policies that can be deployed using different camera poses in the real world. The authors train on 5 tasks, and show success in scene variations on all tasks. They also ablate over different design choices and show that each of them has an important role to play.

## Strength
1. This paper pushes our robot capabilities in the right direction; calibrating camera is an incredible pain point for real robot, so I (and many roboticists) would want this work to succeed.
2. The method is generally sound and the design choices are well motivated and ablated.
3. Experiments over different tasks and many different real scenes show that it is not a fluke or trick but real progress.
4. The qualitative analysis is also interesting and adds a new layer to the understanding.

## Weaknesses
1. **No training curves!** This is an important point to understand if this result scales to for example imitation learning. How many samples does an encoder have to see before it can be completely viewpoint invariant?
2. I am having a hard time imagining how this work can scale to multi-task, language conditioned objectives. The invariance is largely a trick of the encoder, how do we condition our encoder to be multi-task like this?

**Quality Of The Limitations Section:**

3

**Questions For Rebuttal:**

See the weakness section above. The sample complexity of learning such an encoder is of particular interest.

**Robotics Focus:**

4

**Summary Of Paper:**

The authors use sim-to-real with extreme background and scene randomization to train a policy that can be invariant to background and camera pose in practice.

**Summary Of Recommendation:**

This work unlocks new capabilities in robotics and thus should be read by the audience of CoRL with haste.

---

### Official Review · Reviewer_jsuf · 2024-07-20
**Maniwhere: A Reinforcement learning method that can generalize to both changes in camera views and visual appearances**

**Originality:** 4
**Technical Quality:** 4
**Clarity Of Presentation:** 3
**Potential Impact:** 4
**Recommendation:** 4
**Confidence:** 4

**Review:**

### **Quality**

The paper tackles the two most widely faced problems in both reinforcement learning and imitation learning, i.e., changes in visual appearances and camera views. It performs extensive experiments in both simulation and real-world environments. The method translates zero-shot from sim2real.

### **Clarity**

The paper is clearly written, and the contributions are clearly stated and easy to understand.

### **Originality**

The paper proposes a novel approach to learn multi-view representation utilizing one fixed view while varying another view, aiming to bring the learned representations for both views together.

### **Significance of this Work**

The paper makes a significant contribution towards handling variations in camera views and visual perturbations, outperforming all the baselines by 68.5% on average.

### **Strengths**

1. Extensive evaluations on both simulation and real-world tasks.
2. Zero-shot sim2real results: The direct sim2real performance (with no real-world demos) claimed in the paper could be a game-changer for how we deploy RL policies in the real world.

### **Weaknesses**

1. The paper needs to substantiate what components contribute to the handling of visual appearance change. The current presentation addresses and motivates handling multiple camera views, but the paper needs to provide more motivation and qualitative analysis on handling visual diversity. Currently, it's hard to decipher what components contribute to this.
2. In the experiments section, there is no discussion about how much the positions of the object in each task were varied. The policy could just be memorizing to predict the same sequence of actions regardless of the camera views. It is important to stress the generalization of the method to varying object locations.
3. The zero-shot sim2real transfer performance needs to be analyzed both qualitatively and quantitatively. Is the zero-shot transfer a consequence of simple and articulated objects used in the tasks? What happens with more complex tasks with more dynamic contacts where simulation is hard to match the real world?

**Quality Of The Limitations Section:**

2

**Questions For Rebuttal:**

Please refer to the weaknesses section for a list of questions to address for rebuttal.

**Robotics Focus:**

4

**Summary Of Paper:**

The main contributions of the paper include a multi-view representation learning objective to learn invariant features and generalize across time steps, added stn module into the visual encoder (for spatially transforming feature maps), a curriculum of domain randomization to stabilize rl, and extensively evaluate the method on 8 different simulation tasks and their real world clones

**Summary Of Recommendation:**

Maniwhere addresses two crucial issues in robot learning i.e. changes in camera views and visual observations. It could have a significant impact on the training and iteration speed of robot learning algorithms by removing the need for tele-operated demonstrations.

---

### Official Review · Reviewer_RYkn · 2024-07-21
**Initial Review of the Manipulate Anywhere Paper**

**Originality:** 3
**Technical Quality:** 4
**Clarity Of Presentation:** 4
**Potential Impact:** 3
**Recommendation:** 4
**Confidence:** 4

**Review:**

The paper is generally well-written, and the main text or supplementary material clearly explains the details. The proposed framework shows promising results and would potentially be useful to the robotics community.

Strengths:

-	Combines different components into one framework for robust vision-based RL and sim2real transfer
-	Extensive experiments on different robotic tasks using different robotic platforms both in simulation and on the real system
-	Ablation studies that show the necessity of the framework components
-	Significantly better results than baseline methods, which show the potential usefulness of the framework in the robotics community


Weaknesses:

-	Limited novelty as the different components of the framework already exist, but the combination of them in one framework and their application in robotic tasks is new
-	Evaluation only on own tasks and not on known/available benchmarks (especially in simulation), which would lead to more objective comparison with other methods and enhance reproducibility
-	Potentially limited applicability of the STM module to 3D transformations due to transforming projected views of 3D data (see questions below)

**Quality Of The Limitations Section:**

2

**Questions For Rebuttal:**

-	The STN model is extended to include a perspective transformation of the input data. However, the input data is in the form of a projected view in 2D (depth is an additional channel, as explained in the supplementary material). Therefore, it is questionable what kind of 3D transformation is achieved given that the projected view under significantly different camera angles w.r.t. the fixed view would show a completely different part of the robot/environment. Can you elaborate on whether this type of input data is a limitation, and wouldn’t a 3D representation as point clouds (e.g., [1]) allow better spatial transformation and robustness?

-	You mention that MV-MWM requires additional expert demonstrations to train. – Did you actually do this during your training process for the tasks you defined, or did you use a pre-trained MV-MWM model? It is a little unclear to me how the training procedure for MV-MWM was, and additional elaboration on this in the supplementary material will be useful.

-	From the task descriptions and example videos, the tasks look rather simplified w.r.t. spatial relations - does a robotic task always involve very similar spatial relation between objects (e.g., box always placed on the same side of the bowl in approx. the same distance), or you randomize the spatial relations and placements between the objects and object-robot?

-	The viewpoint randomization is described using yaw and pitch angles, which is a little confusing, as they usually are defined w.r.t. the camera coordinate system. I would suggest using azimuth and elevation angles to describe the viewpoint randomization, as they more appropriately refer to positioning the camera around the robot and are defined w.r.t its coordinate system.

-	The limitations section is very general. You only test the robustness of the model to visual disturbances (there are no occlusions or depth sensor noise); it seems that the tasks always involve the same spatial relations between objects or object-robot, and the viewpoint randomization is within a specific range. Please consider improving the limitations section based on the answers to the questions above and these comments.

References:
[1] Wang, Jiayun, Rudrasis Chakraborty, and X. Yu Stella. "Transformer for 3D point clouds." IEEE Transactions on Pattern Analysis and Machine Intelligence 44.8 (2021): 4419-4431.

**Robotics Focus:**

4

**Summary Of Paper:**

The authors propose a framework to train vision-based RL models for robotic manipulation in simulation, where the model is robust to visual disturbances and enables zero-shot sim2real transfer. The approach described combines three components for generalizable manipulation: (1) moving view to canonical view mapping; (2) spatial transformer component and (3) curriculum-staged domain randomization to learn an effective policy in simulation and achieve robust sim2real transfer. The authors train single-task models using this framework for different robotic tasks and different robotic arms and grippers. Several baselines are compared to the proposed method and the evaluation is done both in simulation and on the real systems and the results show that the proposed framework significantly outperforms the baselines.

**Summary Of Recommendation:**

The work tackles relevant topics in robotics (vision-based RL from dynamic views, robust sim2rela transfer) and based on the results as well as the detailed presentation it will be relevant to the robotics community.

---

### Author Rebuttal · Authors · 2024-08-11

We have uploaded the additional experiments requested by the reviewers, along with the results from both simulation and real-world settings. We hope these experiments can address the reviewers' concerns. Additionally, we have also provided relevant result files under each review to facilitate you to locate the corresponding outcomes. We thank all the reviewers for your insightful comments.

---

### Decision · Program_Chairs · 2024-09-04

**Decision:**

Accept

**Comment:**

The reviewers were all impressed by this paper.